# Incidence, Predictors, and Management of Femoral Vascular Complications Following Catheter Ablation for Atrial Fibrillation: A Systematic Duplex Ultrasound Study

**DOI:** 10.3390/biomedicines13020314

**Published:** 2025-01-28

**Authors:** Hyo Jin Lee, Su Hyun Lee, Seongjin Park, Myoung Jung Kim, Juwon Kim, Ju Youn Kim, Seung-Jung Park, Young Keun On, Kyoung-Min Park

**Affiliations:** Division of Cardiology, Department of Internal Medicine, Heart Vascular and Stroke Institute, Samsung Medical Center, Sungkyunkwan University School of Medicine, Seoul 06351, Republic of Korea; hyozhin@naver.com (H.J.L.); juwon1107.kim@samsung.com (J.K.);

**Keywords:** duplex ultrasound, atrial fibrillation, catheter ablation, femoral vein, venipuncture, vascular complication, hematoma, pseudoaneurysm, arteriovenous fistula

## Abstract

**Background/Objectives:** Catheter ablation is an effective treatment for atrial fibrillation (AF) but is associated with femoral vascular complications. While anticoagulation therapy has been linked to these complications, specific risk factors remain unclear. This study assessed the incidence and predictors of vascular complications after catheter ablation using systematic duplex ultrasound (DUS) as well as their outcomes. **Methods:** A single-center observational study was conducted with 404 consecutive AF patients who underwent catheter ablation between March 2023 and February 2024. DUS was performed systematically post-procedure in all patients to identify complications; these were primarily treated with DUS-guided manual compression. **Results:** Vascular complications were observed in 6.4% of patients, higher than reported in previous studies. Hematomas (3.5%) and arteriovenous fistulas (AVFs, 2.0%) were the most common such complications. Multivariate analysis identified repeat ablation (odds ratio [OR] 3.09, 95% confidence interval [CI] 1.10–8.64, *p* = 0.03) and months of experience <6 months (OR 3.42, 95% CI 1.36–8.63, *p* = 0.01) as significant predictors of complications. DUS-guided compression managed most complications successfully, with three pseudoaneurysms resolved through compression and one through embolization. However, AVFs were relatively resistant to conservative management, often necessitating prolonged observation or intervention. **Conclusions:** Systematic DUS following catheter ablation revealed a higher-than-expected incidence of vascular complications. Repeat ablation and months of experience <6 months are potential predictors of femoral vascular complications. DUS-guided compression is effective for most complications, although AVFs present greater treatment challenges.

## 1. Introduction

The catheter ablation is a widely utilized intervention for the treatment of cardiac arrhythmias, including atrial fibrillation (AF). However, this procedure is not always safe [1,2,3]. Complication rates associated with catheter ablation vary depending on the type of procedure [4], patient characteristics, and operator experience [5]. Among these complications, vascular complications are particularly noteworthy due to their frequency and potential severity.

The femoral vein is a common access site for catheter ablation, and the use of ultrasound (US) guidance for femoral venipuncture has been proposed to reduce both major and minor complications associated with the procedure [6]. A meta-analysis indicated that US-guided femoral venipuncture can reduce the incidence of complications such as hematomas, arteriovenous fistulas (AVFs), and pseudoaneurysms [7]. Despite these findings, a randomized controlled trial reported that US-guided femoral venipuncture improved inter-procedural outcomes such as puncture time, number of puncture attempts, inadvertent arterial punctures, and unsuccessful cannulations, but did not significantly decrease the overall rate of major complications [8].

Vascular complications remain the most prevalent type of complication following catheter ablation for AF, with an incidence rate of approximately 1.31% [9]. However, the real incidence of vascular complications, including those that may be clinically silent or “hidden”, is not well understood. Traditional clinical assessments may underestimate the true frequency of these complications, highlighting the need for more sensitive and comprehensive diagnostic tools.

Predictors of vascular complications following catheter ablation are not as well characterized as those for other major complications. Factors such as patient demographics [10], comorbid conditions, procedural specifics, and operator technique may all influence the likelihood of vascular complications. Understanding these predictors is crucial for risk stratification, patient counseling, and preventative strategies.

Given the importance of early detection and management of vascular complications, our study aimed to investigate the incidence of these complications using DUS following catheter ablation. Additionally, we sought to identify predictors of vascular complications to better inform clinical practice and improve patient outcomes. This study addresses a significant gap in the current literature by providing a comprehensive analysis of postoperative vascular complications and their predictors, contributing to the optimization of catheter ablation procedures for AF.

## 2. Materials and Methods

This single-center observational study consisted of 404 consecutive patients with AF who underwent catheter ablation at Samsung Medical Center between March 2023 and February 2024. Detailed patient data and clinical outcomes were obtained from medical records. The hospital’s institutional review board approved the study protocol (IRB No. SMC-2024-01-023-001), which complied with the Declaration of Helsinki.

### 2.1. Procedural Protocols

As recommended in the consensus, anticoagulation therapy in AF ablation was uninterrupted during the periprocedural period [11,12]. The procedure was performed under general anesthesia or sedation with midazolam and propofol. Femoral access was obtained using the modified Seldinger technique, and a 6.5-Fr/7.5-Fr/8.5-Fr-sized sheath for ablation catheter insertion was cannulated. The months of experience were defined as months of venipuncture experience for AF ablation and categorized into less than 6 months and 6 months or more. The operators who attempted vascular puncture were first-year fellows in cardiac electrophysiology and none had prior venipuncture experience for AF catheter ablation. Our institution’s fellowship begins in March, and a new fellow starts performing venipuncture the following year. We defined months of experience < 6 months from early March to the end of August, and months of experience ≥ 6 months from early September to the end of February of the following year. The latest consensus emphasizes the necessity of US-guided venipuncture to reduce vascular complications during catheter ablation of AF, yet 24% among the writing group members still perform venipuncture without US guidance [12]. In most cases, the first attempt used a technique where the femoral artery pulse was palpated, followed by puncturing medially without the aid of US. Venipuncture was performed with the basic principle of puncturing from the lateral to the medial side, without crossing the height of the inguinal crease and without invading the outer boundary where the femoral artery pulse is palpable. There were no additional rules that restricted the operators beyond this.

A transseptal puncture was performed using a Brockenbrough needle (Abbott, St. Paul, MN, USA) and an 8.5-Fr-long sheath (SL-0, St. Jude Medical, Inc., St. Paul, MN, USA). Intracardiac echocardiography was performed during transseptal puncture and ablation procedures for monitoring complications. For reducing complications including cardiac tamponade, intravenous heparin was administered immediately following transseptal puncture during ablation procedures and adjusted to achieve and maintain an activated clotting time (ACT) of at least 300 s [13]. The latest consensus supports initial heparin bolus administration before transseptal puncture to reduce thrombus [12]. Pulmonary vein isolation was performed with either a 28-mm second-generation cryoballoon (Arctic Front Advance, Medtronic, Minneapolis, MN, USA or Polar X, Boston Scientific, St. Paul, MN, USA) via a 15-Fr steerable sheath (FlexCath, Medtronic, Minneapolis, MN, USA or POLARSHEATH, Boston Scientific, St. Paul, MN, USA) or an irrigated-tip radiofrequency catheter (SmartTouch Surround Flow, Biosense Webster, Irvine, CA, USA or FlexAbility, Abbott, St. Paul, MN, USA) via an 8.5-Fr-long sheath (SL-0) guided by a three-dimensional (3D) mapping system (CARTO3, Biosense Webster, Irvine, CA, USA or ENSITE, Abbott, St. Paul, MN, USA). Additional substrate modification was performed according to the operator’s preference.

### 2.2. Postoperative Management

Hemostasis of the vascular access site was achieved using a figure-eight suture after protamine administration. Simple manual compression was used only in cases of failure, and venous vascular closure systems were not utilized. Post-procedure DUS evaluations for all patients that underwent catheter ablation for AF were performed by the vascular surgeons on the day of the procedure or the following day. If symptoms indicating complications, such as pain, swelling, or bleeding, were observed in the patient, DUS was conducted on the same day, otherwise, it was performed the next day. The femoral vascular complications after the procedure were reported to include three types: hematoma of Bleeding Academic Research Consortium (BARC) scale 2 or higher, AVF, and pseudoaneurysm [14]. In cases where complications occurred, most were treated with manual compression hemostasis guided by DUS [15]. For lesions that could not be controlled by compression hemostasis, methods such as embolization were utilized. In instances where embolization was unsuccessful or deemed inappropriate, surgical intervention was considered [16].

### 2.3. Statistical Analysis

Continuous variables are presented as mean ± standard deviation and were compared using Student’s *t*-test for normally distributed data. For categorical variables, frequencies and percentages were calculated, and comparisons between groups were performed using the chi-square test or Fisher’s exact test as appropriate. Variables with a *p*-value of less than 0.3 in the univariate analysis were selected for inclusion in a multivariable logistic regression analysis to identify independent predictors of femoral vascular complications. Multivariable logistic regression was performed using a backward elimination method, and results are expressed as odds ratios (ORs) with corresponding 95% confidence intervals (CIs). Statistical significance was defined as a two-sided *p*-value less than 0.05. All statistical analyses were performed using SPSS software for Windows, version 20.0.0 (IBM Corp., Armonk, NY, USA).

## 3. Results

### 3.1. Baseline Characteristics

In total, 222 (55.0%) and 182 (45.0%) patients underwent cryoablation and radiofrequency catheter ablation for AF, respectively. Baseline characteristics of the 404 patients are shown in Table 1. All patients underwent DUS after catheter ablation for AF before discharge. Among these patients, 378 (93.6%) experienced no complications after catheter ablation. The average age of patients included in the study was 56.8 ± 9.6 years, and there were more male patients (315 men, 78%) than female patients. A total of 395 patients, accounting for 98% of the entire group, was on anticoagulants. Average left atrial size was 43.7 ± 7.0 mm, indicating a relatively enlarged left atrium. Baseline characteristics were not significantly different between the two groups, except for diabetes. Diabetes was not only significantly more prevalent in the group without vascular complications, but it was also found exclusively in that group (*p* < 0.04).

### 3.2. Procedural Characteristics

Table 2 summarizes the procedural characteristics of the study population, categorized by the presence or absence of femoral vascular complications. A history of previous ablation was significantly associated with a higher rate of femoral vascular complications (23.1% vs. 10.1%, *p* = 0.039). However, no significant differences were found between groups regarding the number of venous sheaths used (mean 4.4 ± 0.5 for both groups, *p* = 0.520) or the use of cryoballoon ablation (57.7% vs. 54.8%, *p* = 0.771). Procedure duration was slightly shorter in patients with femoral vascular complications than those without, but this difference was not significant (92.5 ± 23.5 min vs. 102.7 ± 35.9 min, *p* = 0.140).

Additionally, ACT values including minimum (322.5 ± 26.7 s vs. 314.6 ± 29.1 s, respectively, *p* = 0.734), maximum (359.8 ± 31.7 s vs. 360.0 ± 33.2 s, respectively, *p* = 0.763), and mean ACT (335.1 ± 27.7 s vs. 331.0 ± 23.0 s, respectively, *p* = 0.325), showed no significant differences between the groups. The heparin dose administered was similar, with no significant difference observed between patients with and without complications (10,661.5 ± 442.0 IU vs. 10,491.2 ± 146.3 IU, respectively, *p* = 0.361).

### 3.3. Incidence of Femoral Vascular Complications

DUS was performed on all patients either on the day of the procedure or the day after the procedure to diagnose the occurrence of femoral vascular complications. As shown in Figure 1, 26 patients (6.4%) exhibited vascular complications at the femoral access site, including hematomas, AVFs, and pseudoaneurysms in 14 (3.5%), eight (2.0%), and four (1.0%) patients, respectively. DUS, performed on all patients regardless of the presence of symptoms, detected early complications in 24 of 26 patients (92%) with confirmed complications. This study compared patients who underwent radiofrequency ablation (RFCA) and cryoballoon ablation and found no differences in vascular complication rates or outcomes between the two groups. Additionally, hematoma was more common in the cryoballoon ablation group (2.2% vs. 4.5%), while pseudoaneurysm occurred more frequently in the RFCA group (0% vs. 2.2%, Appendix A). However, these differences were not statistically significant (Table 2). As illustrated in Figure 2, the monthly incidence of femoral vascular complications varied throughout the year. Complications were most frequently observed in the early months after the new practitioner began performing venipuncture procedures, with a notable peak in April (five cases), followed by a gradual decrease over the subsequent months.

### 3.4. Management of Femoral Vascular Complications

Management of each complication is detailed in Figure 1. Among hematomas, 93% (13 cases) were observed without any specific intervention. One case of hematoma, which was not detected by DUS but identified through CT angiography, was managed with simple compression without DUS guidance. For AVFs, 38% (3 cases) required additional intervention, with two patients undergoing compression under DUS guidance and one patient receiving embolization. All patients with pseudoaneurysms required intervention; three underwent compression under DUS guidance, and one underwent embolization.

### 3.5. Follow-Up of Femoral Vascular Complications

Figure 3 shows the outcomes of the femoral vascular complications. All patients who developed hematomas or pseudoaneurysms, whether they were observed or received additional interventions, experienced complete resolution of the lesions. Meanwhile, 75% (six cases) of patients who developed AVFs did not achieve resolution, and this group included all three patients who received additional interventions. However, none of the patients in any group required surgical intervention.

### 3.6. Predictors of Vascular Complications

Univariate and multivariate analyses were conducted to identify predictors of femoral vascular complications (Table 3). In the multivariate analysis, repeat ablation remained a significant predictor, with an increased odds of complications (OR 3.09; 95% CI 1.10–8.64; *p* = 0.032). Additionally, months of experience < 6 months showed a significant association with higher complication rates (OR 3.42; 95% CI 1.36–8.63; *p* = 0.009), suggesting a substantial effect of the learning curve on complication risk.

## 4. Discussion

To our knowledge, this study is the first to report the results of systematic conventional venipuncture and post-procedural DUS to identify and manage complications following catheter ablation for AF, including both RFCA and cryoballoon ablation techniques. The main findings of this study are: (1) post-procedural DUS detected a higher incidence of complications than reported in previous studies; (2) repeat ablation and months of experience < 6 months were identified as risk factors for vascular complications; and (3) most vascular complications were managed successfully with DUS-guided compression, though AVFs were relatively challenging to treat.

Ströker E. et al. presented outcomes from one of the three groups in their study, which underwent conventional puncture followed by US evaluation for complications after catheter ablation [17]. Moreover, our findings align with those of prior study, demonstrating that systematic DUS evaluation can detect more complications. However, our study differs from previous research in several important aspects and provides new clinical implications. First, our study is the first to include both cryoballoon ablation and RFCA as catheter ablation techniques in a systematic DUS study. From a population perspective, our study evaluated not only cryoballoon ablation but also RFCA performed with three-dimensional mapping. Importantly, we found no significant differences in vascular complication outcomes between the two techniques (*p* = 0.772). Second, unlike previous research, which did not identify risk factors for vascular complications beyond the absence of US-guided venipuncture, our study identified two additional factors associated with vascular complications: (i) operator-related factors, specifically months of experience <6 months, and (ii) patient-related factors, repeat ablation. Finally, in the previous study, only 265 out of 1435 patients underwent venipuncture without US guidance and were subsequently evaluated by US, with 800 patients not receiving pre- or post-procedure US. In contrast, our study assessed 404 patients who underwent venipuncture without US guidance, with systematic post-procedural US evaluation, highlighting a notable difference in population size.

In our study, vascular complications were observed in 6.4% of patients, a higher incidence than the previously reported rate of approximately 1.31% based on traditional clinical assessment. In another study on cryoballoon ablation for AF, where conventional venipuncture was performed followed by systematic DUS, a major US event rate of 3.8% and a minor US event rate of 7.5% were found, indicating a higher incidence of complications than previously reported [17]. This discrepancy highlights the limitations of conventional methods in detecting “hidden” or subclinical vascular issues. DUS, with its ability to visualize both vascular anatomy and blood flow, was instrumental in identifying complications such as AVFs, pseudoaneurysms, and hematomas that might not have been detected through clinical examination alone. These results reinforce the value of routine postoperative DUS evaluation for early identification and management of vascular complications, which can improve patient outcomes. With the studies of same-day discharge in AF catheter ablation, cardiac implantable electronic device and various procedures requiring venipuncture [18,19,20,21], DUS can help facilitate same-day discharge after the procedures.

An early study comparing conventional venipuncture and US-guided venipuncture in AF ablation showed that US-guided venipuncture was associated with a lower rate of BARC 2+ bleeding (10.4% vs. 19.9%) than conventional venipuncture [22]. A recent study compared vascular complication rates within 30 days between conventional venipuncture and US-guided venipuncture, and reported fewer pseudoaneurysms in the US-guided group. However, there were no statistically significant differences in AV fistula or hematoma incidence requiring treatment between the groups [23]. This study also did not include systematic post-procedural imaging, so the full extent of potential complications could not be captured. The complication rate observed in the conventional venipuncture group in our study was not higher than the rates reported in the US-guided venipuncture groups in these previous studies.

A previous study demonstrated that practitioners who underwent mastery learning with a simulator had a higher success rate in central venous catheterization in the intensive care unit (95% vs. 81%) and a lower incidence of arterial puncture (1% vs. 14%) than those who did not have this training [24]. A relatively recent study found that, while practitioners with one year of training and experience showed an overall reduced complication rate in central vein catheterization compared to their first year with limited experience, a significant reduction was observed only in infection rates [25]. Both of these studies focused on venipuncture for central vein catheterization; no prior studies have compared the learning curve and complication rates specifically for femoral venous access. The most recent prospective multicenter cohort study reported that less experienced practitioners (<100 catheterizations) had a higher incidence of major mechanical complications during US-guided central vein catheterization, including femoral venous access, than more experienced practitioners [26]. This study demonstrated that, even in hospitals where procedures are performed under US guidance, complication rates are higher for practitioners with limited experience.

Our data indicate that procedures performed during the months of experience <6 months were associated with a higher complication risk (Table 3). This heightened risk may be attributed to the practitioner’s limited experience, as the incidence of complications tended to decrease as the practitioner became more skilled over time. Figure 2 supports this finding, showing a peak in complications in the early months following the transition of individual practitioners, particularly in April, when the number of complications was notably high. This recurring pattern suggests that the institutional practice of rotating practitioners annually may inadvertently contribute to increased complication rates each spring. Addressing this learning curve effect through targeted training or extended mentorship could potentially mitigate early complications and enhance patient safety. Further studies to explore structured intervention strategies that reduce the impact of practitioner transitions on complication rates are warranted. Patients who had undergone prior catheter ablation were approximately three times more likely to experience vascular complications, possibly due to cumulative trauma to the femoral vessels or scar tissue formation from prior procedures.

A previous study described a causal relationship between anticoagulation therapy and procedure-related complications [27]. In our study, the association between the total administered dose of anticoagulants and procedure-related complications was not significant. Rather than contradicting previous findings, our study findings suggest that the method of administering anticoagulants and the dosing rate per unit time may be more important than the total amount of anticoagulants used. And recent study has shown an association between age and major complications following venipuncture [28]. However, this study did not reveal such finding, which may be attributed to the lower mean age and narrower standard deviation compared to recent study.

Two patients had complications that could not be diagnosed on DUS; however, due to their symptoms, computed tomography (CT) angiography was performed, which identified a hematoma and a pseudoaneurysm, both requiring interventions. Similar to previous study highlighting the utility of the imaging modality [29], these findings suggest that CT angiography can be beneficial in cases where significant symptoms are present but no abnormalities are detected on DUS, particularly in situations such as deep-seated hemorrhage [30,31].

Figure 4 and Figure 5 illustrate two representative cases of femoral vascular complications: an AVF and a pseudoaneurysm, respectively, highlighting differences in progression and treatment outcomes. Figure 4 depicts a case where an AVF, despite early detection, continued to progress over a five-month period, indicating that conservative management alone may be insufficient for complete resolution in such cases. Differing from previous study [32], this suggests that AVFs may require surgical intervention for effective management due to their potential for prolonged persistence and progression. In contrast, Figure 5 presents a pseudoaneurysm case managed initially with DUS-guided compression. Although the initial compression was unsuccessful, follow-up DUS one month later demonstrated spontaneous thrombotic occlusion and improvement of the lesion. Similar to previous study [33], this finding suggests that pseudoaneurysms may respond positively to conservative management with DUS-guided compression, even if immediate success is not achieved. This trend is consistent with observations from Figure 1 and Figure 2, where all pseudoaneurysm cases demonstrated gradual improvement over time following DUS-guided hemostasis or embolization, regardless of short-term treatment success. Conversely, AVF cases exhibited persistent progression, even when detected early. These results underscore the importance of early detection for both complications, while indicating distinct management approaches: pseudoaneurysms can often be managed conservatively with DUS-guided compression, whereas AVFs may require surgical intervention due to the higher likelihood of treatment resistance and progression, even with early detection. This distinction suggests that tailored management strategies are necessary for femoral vascular complications. For AVF cases, early consideration of surgical intervention may result in better long-term outcomes. These differences emphasize the need for individualized treatment strategies when managing femoral vascular complications.

### Limitations

There are several limitations to our study that warrant discussion. First, as a single-center study with limited populations, the findings may not be generalizable to other institutions with different patient populations or procedural techniques. Second, although the latest consensus recommends US-guided venipuncture, our study included patients who underwent conventional venipuncture due to the resource limitations and needs of our institution. However, 24–33% among the writing group members still perform conventional venipuncture, according to the most recent two consensus statements [12,13]. Third, while we identified significant predictors of vascular complications, our analyses were limited to factors available in the medical records. Other potential predictors, such as genetic predispositions or specific operator techniques, were not examined in detail. Fourth, the relatively short follow-up period may have led to an underestimation of late-onset complications, including those that might manifest weeks after the procedure. Finally, post-procedure management was conducted using the figure-of-eight suture which has been shown to have benefits in several studies as the primary hemostasis method [34,35], despite the proven utility of venous vascular closure systems [36,37,38]. In the future, it will be necessary to investigate vascular complication outcomes in studies utilizing venous vascular closure systems for hemostasis.

## 5. Conclusions

The incidence of vascular complications following catheter ablation for AF is higher than traditionally reported, particularly when sensitive imaging modalities like DUS are used for detection. Repeat ablation and months of experience <6 months were identified as significant predictors of these complications. Most vascular complications were managed successfully with DUS-guided compression, though AVF was relatively challenging to treat.

## Figures and Tables

**Figure 1 biomedicines-13-00314-f001:**
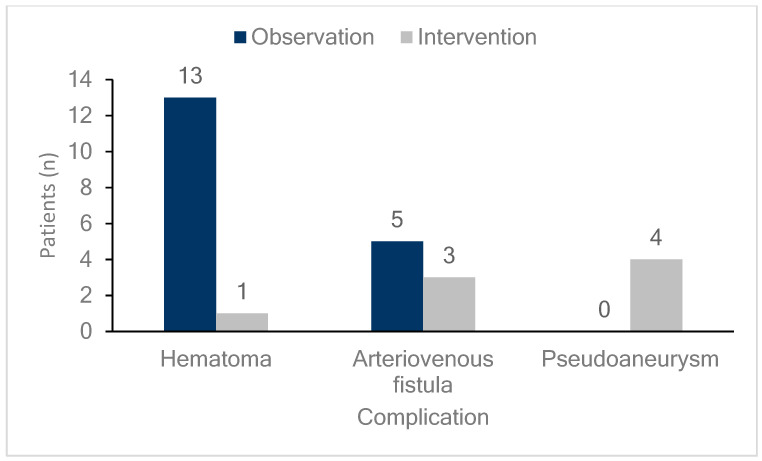
Incidence and management of femoral vascular complications. The bar graph shows the number and percentage of patients experiencing various femoral vascular complications (hematomas, arteriovenous fistulas, and pseudoaneurysms) along with their management strategies. Dark blue bars represent patients managed through observation, while light gray bars indicate those managed with intervention. Hematomas occurred in 14 patients, with 13 (92.9%) managed by observation and one (7.1%) by intervention. Arteriovenous fistulas were observed in eight patients, with five (62.5%) managed by observation and three (37.5%) by intervention. Pseudoaneurysm occurred in four patients, all of whom (100%) were managed with intervention.

**Figure 2 biomedicines-13-00314-f002:**
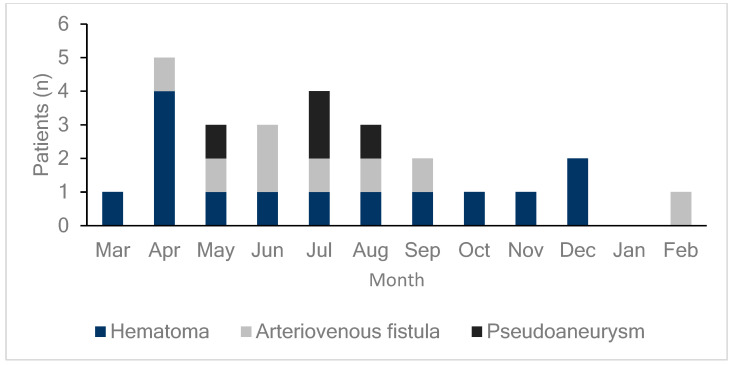
Monthly incidence of femoral vascular complications. This bar graph displays the distribution of femoral vascular complications (hematomas, arteriovenous fistulas, and pseudoaneurysms) by month. Each color represents a specific type of complication: dark blue for hematomas, light gray for arteriovenous fistulas, and black for pseudoaneurysms. Complications peaked in April, with a total of five cases (four hematomas and one arteriovenous fistula), which may reflect the initial learning curve of a new practitioner that started in March. As the new practitioner gains experience over the year, complication rates may decrease, until a new, inexperienced practitioner takes over each March, potentially leading to similar early peaks annually.

**Figure 3 biomedicines-13-00314-f003:**
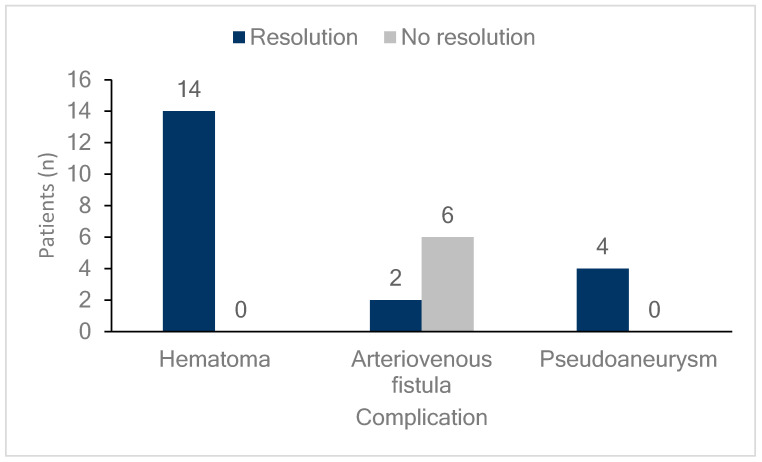
Follow-up of femoral vascular complications. The bar graph illustrates the resolution status of patients with different femoral vascular complications (hematomas, arteriovenous fistulas, and pseudoaneurysms). Dark blue bars indicate patients whose complications resolved, while light gray bars represent those without resolution. Among patients with hematomas, all 14 (100%) showed resolution. For arteriovenous fistulas, two patients (25%) achieved resolution, while six (75%) did not. In cases of pseudoaneurysm, all four patients (100%) experienced resolution.

**Figure 4 biomedicines-13-00314-f004:**
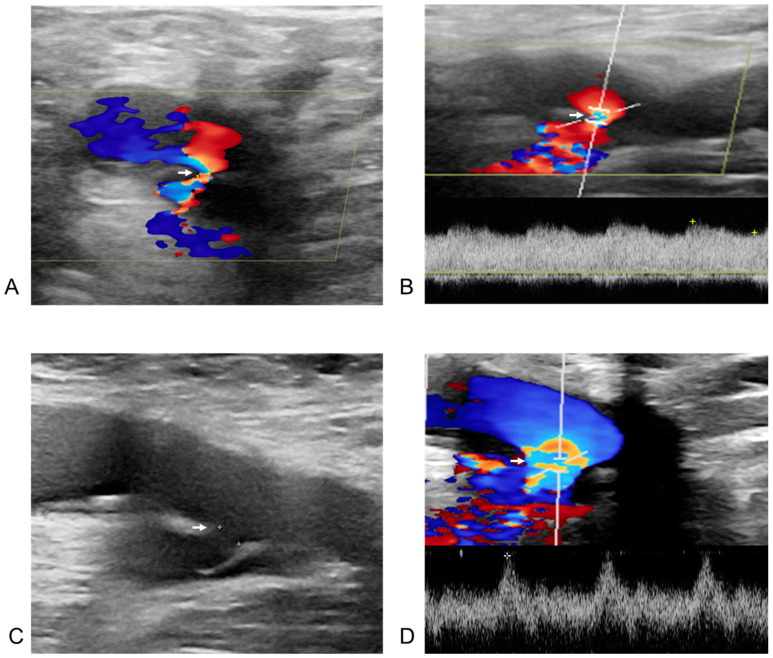
Post-procedural and follow-up duplex ultrasonography (DUS) of an arteriovenous fistula from the right deep femoral artery to the right common femoral vein in a 60-year-old male patient after radiofrequency ablation for atrial fibrillation. The white arrow indicates the neck of the arteriovenous fistula in all images. (**A**) DUS obtained one day after the procedure. The right deep femoral artery is shown above the arrow, and the right common femoral vein below it. Doppler flow acceleration (the colored area indicated by the white arrow) was observed as blood flowed from the right deep femoral artery through the neck into the right common femoral vein. (**B**) Spectral Doppler image from one day post-procedure. The diameter of the neck (the colored area indicated by the white arrow) was measured as 1.0 mm, with a flow velocity of 333.2 cm/s. (**C**) Two-dimensional image at six months after the procedure. The size of the fistula neck (the white arrow) increased from 1.0 mm to 2.6 mm (diameter measured by white spot to spot). (**D**) Spectral Doppler image at six months post-procedure. A more arterialized waveform (the colored area indicated by the white arrow) was observed within the right common femoral vein.

**Figure 5 biomedicines-13-00314-f005:**
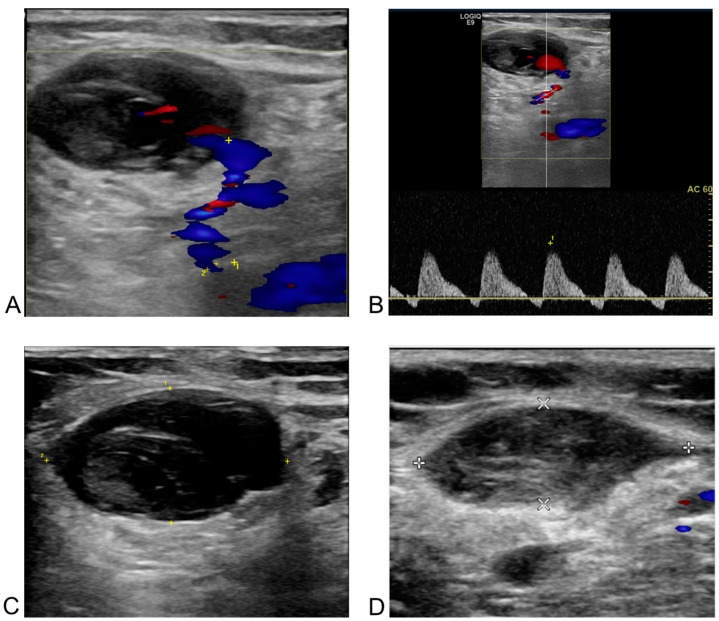
Post-procedural and follow-up duplex ultrasonography (DUS) of a pseudoaneurysm at the left common femoral artery in a 66-year-old female patient after redo-radiofrequency ablation for atrial fibrillation. (**A**) DUS one day after the procedure. The blue area in the bottom right corresponds to the body of the pseudoaneurysm, and the unnumbered cross corresponds to the major axis of the neck. A neck with a 1.4 mm orifice diameter (the distance between the yellow cross labeled 1 and the cross labeled 2) and a length of 16.9 mm (the distance between the two unnumbered yellow crosses) was observed between the left common femoral artery and the pseudoaneurysm on the Doppler image. (**B**) Spectral Doppler image one day after the procedure. “To-and-from” flow was observed with waveforms above and below the baseline in the pseudoaneurysm neck between the arterial vessel and the sac. (**C**) Two-dimensional image taken a day after the procedure. A low echogenicity pseudoaneurysm and its neck are present on the left of the cross-section of the left common femoral artery, though this was challenging to confirm without Doppler imaging. (**D**) Two-dimensional image one month after the procedure. Thrombotic occlusion of the pseudoaneurysm, previously observed in the branch of the common femoral artery in the left inguinal area, was confirmed. The line connecting the two white crosses represents the major axis, and the line connecting the two white Xs represents the minor axis of left common femoral artery.

**Table 1 biomedicines-13-00314-t001:** Baseline characteristics.

	Total	Femoral Vascular Complications		*p* Value *
	(N= 404)	(+) (N= 26)	(−) (N= 378)	
Age, years	56.8 ± 9.6	61.2 ± 10.5	59.7 ± 9.5	0.735
Male, n (%)	315 (78.0%)	22 (84.6%)	293 (77.5%)	0.398
Height, cm	168.4 ± 8.4	170.4 ± 7.3	168.3 ± 8.4	0.281
Weight, kg	72.1 ± 12.1	74.8 ± 11.3	71.9 ± 12.2	0.555
Body mass index, kg/m^2^	25.3 ± 3.3	25.7 ± 3.4	25.3 ± 3.3	0.992
CHA_2_DS_2_-VASc score	1.5 ± 1.3	1.5 ± 1.3	1.5 ± 1.3	0.960
Hypertension, n (%)	170 (42.1%)	13 (50.0%)	157 (41.5%)	0.398
Diabetes, n (%)	52 (12.9%)	0 (0.0%)	52 (13.8%)	0.043
Medications, n (%)				
Aspirin	3 (0.7%)	0 (0.0%)	3 (0.8%)	0.648
P2Y12 inhibitor	10 (2.5%)	0 (0.0%)	10 (2.6%)	0.401
Dabigatran	5 (1.2%)	1 (3.8%)	4 (1.1%)	0.214
Rivaroxaban	113 (28.0%)	10 (38.5%)	103 (27.2%)	0.218
Apixaban	92 (22.8%)	5 (19.2%)	87 (23.0%)	0.656
Edoxaban	181 (44.8%)	10 (38.5%)	171 (45.2%)	0.502
Warfarin	9 (2.2%)	0 (0.0%)	9 (2.4%)	0.426
Echocardiogram				
LV EF, %	59.6 ± 9.0	63.3 ± 5.3	59.4 ± 9.1	0.137
LA dimension, mm	43.7 ± 7.0	42.5 ± 6.3	43.7 ± 7.0	0.201
E/e’	9.4 ± 4.4	8.1 ± 2.6	9.5 ± 4.5	0.259
Laboratory findings				
Creatinine, mg/dL	0.92 ± 0.48	0.87 ± 0.13	0.93 ± 0.50	0.467
Platelet, 10^3^/uL	214.9 ± 51.6	200.2 ± 38.1	215.9 ± 52.3	0.217
INR	1.41 ± 0.94	1.39 ± 0.26	1.41 ± 0.97	0.554

* Continuous variables were compared using Student’s *t*-test. Categorical variables were compared using the chi-square test or Fisher’s exact test as appropriate. Statistical significance was defined as a two-sided *p*-value less than 0.05. CHA2DS2-VASc score, congestive heart failure, hypertension, age ≥ 75 years (doubled), diabetes mellitus, prior stroke or TIA or thromboembolism (doubled), vascular disease, age 65 to 74 years, sex category; E/e’, ratio of mitral peak E velocity to tissue Doppler early diastolic velocity e’; INR, international normalized ratio; LA, left atrium; LV EF, left ventricle ejection fraction.

**Table 2 biomedicines-13-00314-t002:** Procedural characteristics.

	Total	Complication (+)	Complication (−)	*p* Value *
	(n = 404)	(n = 26)	(n = 378)	
Repeat ablation, n (%)	44 (10.9%)	5 (23.1%)	38 (10.1%)	0.039
Number of venous sheaths, n	4.4 ± 0.5	4.4 ± 0.5	4.4 ± 0.5	0.520
Cryoballoon ablation, n (%)	222 (55.0%)	15 (57.7%)	207 (54.8%)	0.771
Procedure duration, min	102.0 ± 35.3	92.5 ± 23.5	102.7 ± 35.9	0.140
ACT min, s	315.1 ± 29.0	322.5 ± 26.7	314.6 ± 29.1	0.734
ACT max, s	360.0 ± 33.1	359.8 ± 31.7	360.0 ± 33.2	0.763
ACT mean, s	331.3 ± 23.3	335.1 ± 27.7	331.0 ± 23.0	0.325
Heparin dose, IU	10,502.2 ± 2807.9	10,661.5 ± 442.0	10,491.2 ± 146.3	0.361

* Continuous variables were compared using Student’s *t*-test. Statistical significance was defined as a two-sided *p*-value less than 0.05. ACT, activated clotting time; IU, international unit.

**Table 3 biomedicines-13-00314-t003:** Predictors of vascular complications.

Variable	Univariate Analysis			Multivariate Analysis		
	Odds Ratio	95% CI	*p* Value	Odds Ratio	95% CI	*p* Value *
Age	1.02	0.97–1.06	0.461			
Male	1.60	0.54–4.76	0.402			
Height	1.03	0.98–1.08	0.220	1.04	0.97–1.11	0.312
Weight	1.02	0.99–1.05	0.253	1.01	0.97–1.05	0.716
BMI	1.04	0.92–1.17	0.530			
CHA_2_DS_2_-VASc score	0.98	0.72–1.33	0.905			
Hypertension	1.41	0.64–3.12	0.400			
INR	0.98	0.61–1.56	0.928			
Repeat ablation	2.68	1.02–7.10	0.046	3.09	1.10–8.64	0.032
Cryoballoon ablation	1.13	0.50–2.52	0.772			
Procedure time	0.99	0.98–1.00	0.156	0.99	0.98–1.01	0.221
Months of experience < 6 months	3.02	1.24–7.35	0.015	3.42	1.36–8.63	0.009

* Variables with a *p*-value of less than 0.3 in the univariate analysis were selected for inclusion in a multivariate analysis. Statistical significance was defined as a *p*-value less than 0.05. AF, atrial fibrillation; BMI, body mass index; CHA2DS2-VASc score, congestive heart failure, hypertension, Age ≥ 75 years (doubled), diabetes mellitus, prior stroke or TIA or thromboembolism (doubled), vascular disease, Age 65 to 74 years, Sex category; E/e’, ratio of mitral peak E velocity to tissue Doppler early diastolic velocity e’; INR, international normalized ratio.

## Data Availability

The data presented in this study are available on request from the corresponding author.

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
