# Peer review of "Incidence, Predictors, and Management of Femoral Vascular Complications Following Catheter Ablation for Atrial Fibrillation: A Systematic Duplex Ultrasound Study"

_biomedicines, 2025, doi:10.3390/biomedicines13020314_

Round 1

Reviewer 1 Report

Comments and Suggestions for Authors

Great work! Congratulations! Your work has examined the vascular complications that may arise after catheter ablation performed in patients with atrial fibrillation. The work was conducted on a large number of patients (404) over a period of one year in order to highlight the incidence of vascular complications that was 6.4%, higher than previous studies, with the presence of hematomas (3.5%), arteriovenous fistulas (AVF, 2.0%) and pseudoaneurysms (1.0%). Most of the complications were managed with manual compression guided by duplex ultrasound (DUS). Hematomas were mainly observed without specific interventions, while AVFs and pseudoaneurysms required more complex interventions, including embolization. Previous experience of ablation for atrial fibrillation and operator experience in the first six months were identified as significant predictors of vascular complications. Statistical analysis was well conducted: univariate and multivariate analysis were used to identify predictors of femoral vascular complications after catheter ablation for atrial fibrillation. Variables analyzed in the univariate and multivariate analysis to identify predictors of femoral vascular complications after catheter ablation for atrial fibrillation included age, sex, height, weight, BMI, CHA2DS2-VASc score, hypertension, INR, history of previous ablations for AF, cryoballoon ablation, duration of procedure, and procedures during the first six months of operator experience. The results obtained with the univariate analysis demonstrated statistical significance for both the history of previous AF ablations (Odds ratio (OR) 2.68; p = 0.046) and for procedures during the first six months of operator experience (OR 3.02; p = 0.015). The multivariate analysis also showed statistical significance for the history of previous AF ablations (OR 3.09; p = 0.032) and for procedures during the first six months of operator experience (OR 3.42; p = 0.009). These results indicate that a history of previous AF ablations and operator experience in the first six months are significant predictors of femoral vascular complications. These results indicate that weight is not a significant predictor of femoral vascular complications in the multivariate analysis, as the p value is greater than 0.05. With your work you have demonstrated that the systematic use of DUS is able to detect a higher incidence of complications than reported in the literature to date. DUS-guided compression was effective for the majority of complications, although AVFs present greater challenges in treatment. Your study has therefore highlighted the importance of DUS in the early detection and management of post-ablation vascular complications, thus improving patient outcomes. My only criticism: I would have expanded the bibliography a little more!

Author Response

1. Summary

Thank you sincerely for your review and advice. We have added seven references.

2. Point-by-point response to Comments and Suggestions for Authors

Comments 1: I would have expanded the bibliography a little more!

Response 1: We agree with this comment. Therefore, we have added seven references.

Reviewer 2 Report

Comments and Suggestions for Authors Dear Authors,

I read with interest your paper titled: "Incidence, predictors, and management of femoral vascular complications following catheter ablation for atrial fibrillation: A systematic duplex ultrasound study."

You conducted a single-center study analyzing the incidence and predictors of vascular complications after catheter ablation, using systematic duplex ultrasound (DUS) assessment, as well as their outcomes. Your findings highlighted a vascular complication rate of 6.4%, with the majority being hematomas (3.5%) and arteriovenous fistulas (2.0%). Additionally, DUS-guided compression was shown to be effective, with three pseudoaneurysms successfully resolved via compression and one through embolization.

Your multivariate analysis identified prior experience with atrial fibrillation (AF) ablation (odds ratio [OR] 3.09, 95% confidence interval [CI] 1.10–8.64, p = 0.03) and procedures conducted during operators’ first six months of practice (OR 3.42, 95% CI 1.36–8.63, p = 0.01) as significant predictors of complications.

However, I would like to raise a few points for consideration:

  1. Systematic Use of Post-Procedural Ultrasound:
    You correctly noted that the observation of higher vascular complication rates with systematic ultrasound assessments the day after the procedure has been previously reported (e.g., Ströker et al., "Value of ultrasound for access guidance and detection of subclinical vascular complications in the setting of atrial fibrillation cryoballoon ablation," DOI: 10.1093/europace/euy154). While your findings corroborate existing knowledge, they may not significantly advance the field.

  2. Lack of Ultrasound-Guided Venipuncture:
    A notable limitation of your study is the absence of ultrasound-guided femoral venipuncture. While you acknowledged in your discussion (lines 236–239) that this technique improves procedural metrics (e.g., reducing puncture attempts, inadvertent arterial punctures, and puncture time), you suggested its impact on major vascular complications remains inconclusive. I respectfully disagree. In my experience, as well as that of other groups, ultrasound-guided venipuncture is highly effective in reducing vascular complication rates, especially in the context of uninterrupted anticoagulation therapy for AF ablation. For example, in the study by Ströker et al., the complication rate in the ultrasound-guided cohort was 0%.

  3. Guidelines and Recommendations:
    The 2024 European Heart Rhythm Association/Heart Rhythm Society/Asia Pacific Heart Rhythm Society/Latin American Heart Rhythm Society expert consensus statement on catheter and surgical ablation of atrial fibrillation explicitly recommends implementing ultrasound-guided venipuncture routinely to minimize vascular complications. This omission in your methodology diminishes the practical applicability of your findings.

Conclusion:
While your study provides valuable data on vascular complications and their management, it does not offer substantial novel insights into the field. The lack of ultrasound-guided venipuncture, despite its proven efficacy, is a significant limitation that should be addressed in future studies.

Thank you for your contribution to this important topic. I look forward to seeing further advancements in this area.

Author Response

1. Summary

We appreciate editor and reviewers for valuable comments.

We fully agree with the comments of Reviewer 2, and we have submitted our responses as outlined below. Through Reviewer 2's comments, we clarified the distinctions between our study and previous research in the Discussion section, acknowledged the benefits of ultrasound-guided venipuncture in reducing vascular complications, and detailed the reasons and limitations of employing conventional venipuncture in our practice in the Methods and Limitations sections, despite recent consensus recommendations. We sincerely appreciate the constructive comments from the Reviewer 2, and we are ready to respond promptly and accurately to any further requests.

2. Point-by-point response to Comments and Suggestions for Authors

Comments 1: Systematic Use of Post-Procedural Ultrasound:

You correctly noted that the observation of higher vascular complication rates with systematic ultrasound assessments the day after the procedure has been previously reported (e.g., Ströker et al., "Value of ultrasound for access guidance and detection of subclinical vascular complications in the setting of atrial fibrillation cryoballoon ablation," DOI: 10.1093/europace/euy154). While your findings corroborate existing knowledge, they may not significantly advance the field.

Response 1: We agree with this comment. We added the discussion related clinical implication compared with previous study as revised manuscript Line 242 in Page 8.

Comments 2: Lack of Ultrasound-Guided Venipuncture:

A notable limitation of your study is the absence of ultrasound-guided femoral venipuncture. While you acknowledged in your discussion (lines 236–239) that this technique improves procedural metrics (e.g., reducing puncture attempts, inadvertent arterial punctures, and puncture time), you suggested its impact on major vascular complications remains inconclusive. I respectfully disagree. In my experience, as well as that of other groups, ultrasound-guided venipuncture is highly effective in reducing vascular complication rates, especially in the context of uninterrupted anticoagulation therapy for AF ablation. For example, in the study by Ströker et al., the complication rate in the ultrasound-guided cohort was 0%.

Response 2: We agree with your valuable comment. We removed the statement in the discussion regarding the inconclusive outcomes of ultrasound-guided venipuncture for vascular complications and addressed the lack of ultrasound-guided venipuncture as a limitation in our study, and have added the limitation as revised manuscript Line 395 in Page 13.

Comments 3: Guidelines and Recommendations:

The 2024 European Heart Rhythm Association/Heart Rhythm Society/Asia Pacific Heart Rhythm Society/Latin American Heart Rhythm Society expert consensus statement on catheter and surgical ablation of atrial fibrillation explicitly recommends implementing ultrasound-guided venipuncture routinely to minimize vascular complications. This omission in your methodology diminishes the practical applicability of your findings.

Response 3: Thank you for your insightful comment. 24-33% among the writing group members still perform conventional venipuncture, according to the most recent two consensus statements. Therefore, we described the practical realities of performing under such circumstances in the methods section and reiterated this point in the limitations section. And we have added the method and limitation as revised manuscript Line 87 in Page 2, Line 395 in Page 13, respectively.

Reviewer 3 Report

Comments and Suggestions for Authors

The absence of stratification based on procedural types (e.g., cryoballoon versus radiofrequency) in the methodological design could potentially miss variations in complications that are specific to each technique.

The procedural protocols would be enhanced by providing more comprehensive descriptions of the operators' techniques, especially concerning venous puncture methods.

Elaborate on the reasoning behind the chosen anticoagulation protocol, and compare it with alternative approaches recommended in current guidelines.

Provide more detailed information on how operator experience was defined and measured.

A power analysis for the dataset is lacking, leaving it uncertain whether the sample size of 404 patients was adequate to detect the effect sizes reported.

The discussion could be strengthened by offering a more in-depth comparison with existing studies, particularly regarding the effectiveness of DUS relative to other imaging methods.  Furthermore, the authors should highlight in their discussion the potential for same-day discharge, facilitated by the low incidence of vascular access complications, as observed in other areas of arrhythmology (doi: 10.1111/jce.16147)

Author Response

  1. Summary

We appreciate editor and reviewers for valuable comments.

We fully agree with the comments of Reviewer 3, and we have submitted our responses as outlined below.

  1. Point-by-point response to Comments and Suggestions for Authors

Comments 1: The absence of stratification based on procedural types (e.g., cryoballoon versus radiofrequency) in the methodological design could potentially miss variations in complications that are specific to each technique.

Response 1: We agree with this comment. We have added the results with Supplementary figure 1. as revised manuscript Line 181 in Page 5.

Comments 2: The procedural protocols would be enhanced by providing more comprehensive descriptions of the operators' techniques, especially concerning venous puncture methods.

Response 2: We agree with your valuable comment. We have added the description of the venipuncture technique in the method section as revised manuscript Line 91 in Page 3.

Comments 3: Elaborate on the reasoning behind the chosen anticoagulation protocol, and compare it with alternative approaches recommended in current guidelines.

Response 3: Thank you for your insightful comment. We have added the chosen anticoagulation protocol (For reducing complications including cardiac tamponade, intravenous heparin was administered immediately following transseptal puncture in our institution) and an alternative approach of latest consensus in the method section as revised manuscript Line 99 in Page 3.

Comments 4: Provide more detailed information on how operator experience was defined and measured.

Response 4: Thank you for your detailed comment. The months of experience defined as months of venipuncture experience for AF ablation and categorized into less than 6 months and 6 months or more. The operators who attempted vascular puncture were first-year fellows in cardiac electrophysiology and none had prior venipuncture experience for AF catheter ablation. Our institution's fellowship begins in March, and a new fellow starts performing venipuncture the following year. We defined months of experience <6 months from early March to the end of August, and months of experience ≥6 months from early September to the end of February of the following year. And we have added the definition of months of experience in the method section as revised manuscript Line 80, Page 2.

Comments 5: A power analysis for the dataset is lacking, leaving it uncertain whether the sample size of 404 patients was adequate to detect the effect sizes reported.

Response 5: We agree with this opinion. We have added the limitation as revised manuscript Line 393 in Page 13. And we emphasize that in the previous study, only 265 out of 1,435 patients underwent venipuncture without USG guidance and were subsequently evaluated by US, with 800 patients not receiving pre- or post-procedure US (revised manuscript, Line 256 in Page 8).

Comments 6: The discussion could be strengthened by offering a more in-depth comparison with existing studies, particularly regarding the effectiveness of DUS relative to other imaging methods.  Furthermore, the authors should highlight in their discussion the potential for same-day discharge, facilitated by the low incidence of vascular access complications, as observed in other areas of arrhythmology (doi: 10.1111/jce.16147)

Response 6: Thank you for your valuable comment. We have added the discussion as revised manuscript Line 340 in Page 10, Line 272 in Page 8, respectively.

Round 2

Reviewer 2 Report

Comments and Suggestions for Authors

Dear Authors,

I have reviewed the revised version of your manuscript and am pleased with how you addressed the points raised in my previous review.

I acknowledge that some groups continue to rely on conventional venipuncture. One of the strengths of your work is the emphasis on promptly recognizing complications through the systematic use of Doppler ultrasound the day after the procedure.

I hope that, in the near future, you will consider incorporating ultrasound-guided venipuncture into your workflow.

Thank you again for the opportunity to review your valuable work.

Reviewer 3 Report

Comments and Suggestions for Authors

Authors did a remarkable job in answering all of my comments.